# Meaningful changes in motor function in Duchenne muscular dystrophy (DMD): A multi-center study

Francesco Muntoni[1], James Signorovitch[2,3]*, Gautam Sajeev[2], Nicolae Done[2], Zhiwen Yao[2], Nathalie Goemans[4], Craig McDonald[5], Eugenio Mercuri[6], Erik H. Niks[7], Brenda Wong[8], Krista Vandenborne[9], Volker Straub[10], Imelda J. M. de Groot[11], Cuixia Tian[12], Adnan Manzur[1], Ibrahima Dieye[2], Henry Lane[2], Susan J. Ward[3], Laurent Servais[13,14], PRO-DMD-01 study investigators[¶], Association Française contre les Myopathies[¶], The UK NorthStar Clinical Network[¶], ImagingDMD investigators[¶], cTAP[¶]

1 Dubowitz Neuromuscular Centre, NIHR Great Ormond Street Hospital Biomedical Research Centre, Great Ormond Street Institute of Child Health, University College London, & Great Ormond Street Hospital Trust, London, United Kingdom, 2 Analysis Group, Inc., Boston, Massachusetts, United States of America, 3 The collaborative Trajectory Analysis Project, Cambridge, Massachusetts, United States of America, 4 Child Neurology, University Hospitals Leuven, Leuven, Belgium, 5 Department of Physical Medicine and Rehabilitation and Pediatrics, University of California, Davis, Sacramento, California, United States of America, 6 Department of Pediatric Neurology, Fondazione Policlinico Gemelli IRCCS, Catholic University, Rome, Italy, 7 Department of Neurology, Leiden University Medical Centre, Leiden, Netherlands, 8 Department of Pediatrics, University of Massachusetts Medical School, Worcester, Massachusetts, United States of America, 9 Department of Physical Therapy, University of Florida, Gainesville, Florida, United States of America, 10 John Walton Muscular Dystrophy Research Centre, Newcastle University and Newcastle Hospitals NHS Foundation Trust, Newcastle upon Tyne, United Kingdom, 11 Department of Rehabilitation, Donders Centre of Neuroscience, Radboud University Nijmegen Medical Center, Nijmegen, Netherlands, 12 Cincinnati Children's Hospital Medical Center, Cincinnati, Ohio & College of Medicine, University of Cincinnati, Cincinnati, Ohio, United States of America, 13 Department of Paediatrics, MDUK Oxford Neuromuscular Center, University of Oxford, Oxford, United Kingdom, 14 Neuromuscular Center of Liège, Division of Paediatrics, CHU and University of Liège, Liège, Belgium

¶ Memberships are provided in the Acknowledgments
* James.signorovitch@analysisgroup.com

**Data Availability Statement:** All relevant aggregate data are reported within the manuscript and the Supporting Information files. This study uses third party data sources accessed by the collaborative

## Abstract

Evaluations of treatment efficacy in Duchenne muscular dystrophy (DMD), a rare genetic disease that results in progressive muscle wasting, require an understanding of the 'meaningfulness' of changes in functional measures. We estimated the minimal detectable change (MDC) for selected motor function measures in ambulatory DMD, i.e., the minimal degree of measured change needed to be confident that true underlying change has occurred rather than transient variation or measurement error. MDC estimates were compared across multiple data sources, representing >1000 DMD patients in clinical trials and real-world clinical practice settings. Included patients were ambulatory, aged ≥4 to <18 years and receiving steroids. Minimal clinically important differences (MCIDs) for worsening were also estimated. Estimated MDC thresholds for >80% confidence in true change were 2.8 units for the North Star Ambulatory Assessment (NSAA) total score, 1.3 seconds for the 4-stair climb (4SC) completion time, 0.36 stairs/second for 4SC velocity and 36.3 meters for the 6-minute walk distance (6MWD). MDC estimates were similar across clinical trial and real-world data sources, and tended to be slightly larger than MCIDs for these measures. The identified

Trajectory Analysis Project (cTAP), through data use agreements with the relevant data holders. Individual-level data for some of these data sources are available on public repositories, and can be accessed through requests to the data holders via these repositories. • PRO-DMD-01 study and drisapersen trials: Subject-level data from the PRO-DMD-01 study and drisapersen clinical trials were provided by CureDuchenne. Data for these studies can be requested through Vivli (https://www.vivli.org). o PRO-DMD-01 study: https://doi.org/10.25934/00005087 o Drisapersen phase 2 trials: https://doi.org/10.25934/00005153 and https://doi.org/10.25934/00005154 o Drisapersen phase 3 trial: https://doi.org/10.25934/00005155 • Tadalafil DMD trial: https://doi.org/10.25934/00004256 Other data sources used are not available on public repositories and may be available via data use agreements with the individual data holders. Requests for individual patient data may be directed towards the individual institutions/organizations that have collected and curated these patient data. These organizations will consider data requests according to their own data sharing policies and governance: • Ataluren trials and Marathon clinical trial data: Please contact PTC Therapeutics (www.ptcbio.com) • UZ Leuven data: UZ Leuven Data Access Committee (dac@uzleuven.be) • iMDEX study data: Francesco Muntoni, Primary Investigator (f.muntoni@ucl.ac.uk) • North Star UK network data: Salma Samsuddin, Research Coordinator (s.samsuddin@ucl.ac.uk) • CCHMC data: Cuixia Tian, Primary Investigator (cuixia.tian@cchmc.org) • ImagingDMD study data: Krista Vandenborne, Director of ImagingDMD, (kvandenb@phhp.ufl.edu).

**Funding:** This study was conducted by the collaborative Trajectory Analysis Project (cTAP), a precompetitive coalition of academic clinicians, drug developers, and patient foundations formed in 2015 to overcome the challenges of high variation in clinical trials in DMD. cTAP has received sponsorship from Astellas (Mitobridge), Avidity Biosciences, BioMarin Pharmaceutical, Bristol Meyers Squibb, Catabasis, Daiichi Sankyo, Edgewise Therapeutics, Entrada Therapeutics, FibroGen, Italfarmaco SpA, Marathon Pharmaceuticals, NS Pharma, Pfizer, PTC Therapeutics, Roche, Sarepta Therapeutics, Shire, Solid Biosciences, Summit Therapeutics, Ultragenyx, Vertex Pharmaceuticals, Parent Project Muscular Dystrophy, Charley's Fund, and CureDuchenne, a founding patient advocacy partner and provider of initial seed funding to cTAP. Physical function testing at Universitaire Ziekenhuizen Leuven was funded by Fonds

thresholds can be used to inform endpoint definitions, or as benchmarks for monitoring individual changes in motor function in ambulatory DMD.

## Introduction

Progressive deficits in motor function are among the first signs and symptoms of Duchenne muscular dystrophy (DMD), an X-linked recessive disease occurring in 4.78 per 100,000 males [1,2], and typically result in loss of independent ambulation by the early teens [3]. Clinical measurements of ambulatory motor function, including the North Star Ambulatory Assessment (NSAA) [4,5], the six-minute walk distance (6MWD) and the timed 4-stair-climb (4SC), have helped characterize disease progression in DMD [6–10] and have served as primary endpoints in clinical trials of investigational therapies [8,11,12]. These measures change over time for individuals with DMD due to (1) the cumulative impacts of disease progression and maturation; (2) the transient impacts of fatigue, motivation, minor injury, adaptation or other biological factors that temporarily impact a patient's ambulatory performance; and (3) measurement error due to assessor effects, such as varying levels of encouragement provided, small errors in timing, or subjectivity in functional scoring [13].

The present study estimates the minimal detectable change (MDC) in these measures of ambulatory function in DMD–that is, the minimal level of change in *measured* function that must be observed to be reasonably confident that *true, underlying change* in function has occurred for an individual (per item 1 above) rather than only transient changes (per items 2 and 3 above) [14,15]. Understanding MDCs is important to the design and interpretation of clinical studies, and for monitoring real-world outcomes. Definitions of disease progression for use in clinical trial endpoints or treatment algorithms, for example, should be based on MDCs to ensure that signal for progression is distinguished from biological and measurement noise at the individual-patient level. To complement the MDCs, the present study also estimates minimal clinically important differences (MCIDs) in motor function, i.e., differences in function perceived as meaningful to patients [14–17].

MDCs and MCIDs represent different and complementary properties of a functional measure [14,16,18–20]. When the MDC is smaller than the MCID, sub-clinical levels of progression are measurable, and may be important to monitor in the context of a progressive disease despite having no immediate impact on how the patient feels or functions. In contrast, when the MDC is larger than the MCID, some fluctuations in measured function are meaningful to patients but cannot be attributed with an acceptable level of evidence to underlying disease progression or improvement. Sentence intelligibility in amyotrophic lateral sclerosis, for example, has an MDC that is over eight times larger than the MCID [21].

In DMD, MCIDs have been estimated for various measures of motor function: the 6MWD, the NSAA total score, a transformed version of the NSAA, and timed function tests including the 4SC [7,22–24]. However, there have been few assessments of the MDC, i.e., the degree of change measured for an individual that can be confidently interpreted as true functional change, or comparisons of these metrics across real-world and clinical trial settings. In the present study we estimated MDCs based on longitudinal assessments of 6MWD, NSAA and 4SC across multiple clinical centers, networks and clinical trial placebo arms. We assessed the sensitivity of MDCs to data source, patient age and level of motor function. Anchor-based MCIDs for worsening were also estimated as a reference point.

Spierzieke Kinderen (https://www.spierfonds.nl/spierziekten). The PRO-DMD-01 study was sponsored by BioMarin Pharmaceuticals (https://www.biomarin.com/) and data were provided to cTAP by CureDuchenne. The study sponsors were involved in several aspects of the research, including the study design, interpretation of data, and review of the manuscript. The final decision to submit the manuscript for publication was the sole decision of the authors.

**Competing interests:** We have read the journal's policy and the authors of this manuscript have the following competing interests: Francesco Muntoni is a member of the Rare Disease Scientific Advisory Group for Pfizer and of Dyne Therapeutics SAB, and has participated to SAB meetings for PTC, Sarepta, Santhera, Wave Therapeutics. UCL and Great Ormond Street Hospital are recipient of grants from Pfizer, Italfarmaco, Wave, Santhera, Sarepta regarding clinical trials. James Signorovitch co-founded the collaborative Trajectory Analysis Project (cTAP) and is an employee of Analysis Group, Inc., a consulting firm that received funding from the membership of cTAP to conduct this study. Gautam Sajeev, Nicolae Done, and Henry Lane are employees of Analysis Group, Inc., a consulting firm that received funding from the membership of cTAP to conduct this study. Ibrahima Dieye and Zhiwen Yao were employees of Analysis Group, Inc. at the time this study was conducted. Nathalie Goemans has served on clinical steering committees and/or as a consultant and received compensation from Eli Lilly, Italfarmaco, PTC Therapeutics, and BioMarin Pharmaceutical; has served as site investigator for GlaxoSmithKline, Prosensa, BioMarin Pharmaceutical, Italfarmaco, Roche, and Eli Lilly. Craig McDonald has served as a consultant for PTC Therapeutics, BioMarin Pharmaceutical, Sarepta Therapeutics, Eli Lilly, Pfizer Inc, Santhera Pharmaceuticals, Cardero Therapeutics, Inc, Catabasis Pharmaceuticals, Capricor Therapeutics, Astellas Pharma (Mitobridge), and FibroGen, Inc; serves on external advisory boards related to DMD for PTC Therapeutics, Sarepta Therapeutics, Santhera Pharmaceuticals, and Capricor Therapeutics; and reports grants from US Department of Education/National Institute on Disability and Rehabilitation Research, the National Institute on Disability, Independent Living, and Rehabilitation Research, US NIH/National Institute of Arthritis and Musculoskeletal and Skin Diseases, NIH/National Institute of Neurologic Disorders and Stroke, US Department of Defense, and Parent Project Muscular Dystrophy US. Eugenio Mercuri has served on clinical steering committees and/or as a consultant for Eli Lilly, Italfarmaco, PTC

## Methods

### Data sources

Patient-level data from eight clinical trial arms and six real-world data (RWD) or natural history data (NHD) sources accessed by the collaborative Trajectory Analysis Project (cTAP) in 2020–2021 were used in this study (**S1 and S2 Tables**). Studied clinical trial placebo arms included the phase 3 trial of tadalafil [25] (provided by Lilly), the phase 2b and phase 3 trials of ataluren [26,27] (provided by PTC Therapeutics) and two phase 2 trials and one phase 3 trial of drisapersen [28,29] (provided by CureDuchenne). Data were also available from two trials of deflazacort versus prednisone [30] (provided by Marathon/PTC). Curated RWD from DMD clinics was provided by Universitaire Ziekenhuizen Leuven (Leuven), Cincinnati Children's Hospital Medical Center (CCHMC) and the North Star UK database (http://www.northstardmd.com) (NSUK). Collaborators sharing NHD were the iMDEX study (provided by University College London on behalf of the Association Française contre les Myopathies [AFM]), the ImagingDMD study (provided by the University of Florida) and the PRO-DMD-01 prospective natural history study (provided by CureDuchenne). Functional assessments available in these data sources varied as summarized in **S3 Table**.

Clinical co-authors of this manuscript who cared for patients at the included RWD/NHD sources may have access to information that could identify individual participants during or after data collection. For all data sources, only pre-existing, anonymous, de-identified data were analyzed, and informed consent/assent was obtained as needed by data providers. RWD/NHD sources were approved by ethics committees from each contributing institution (the University Hospitals Leuven [Leuven], each participating center of the PRO-DMD-01 study and ImagingDMD, and the institutional review board at the CCHMC (IRB #2010–1881). For the iMDEX study, ethics review boards at the participating institutions approved the study protocol, consent and assent documents. For use of the North Star UK data, this project followed Caldicott Guardian regulations and information was entered in the database after written informed consent was obtained from patients' parents. All clinical investigations were conducted according to the principles expressed in the Declaration of Helsinki.

### Study measures

Meaningful changes were studied for the following motor function measures: 1) NSAA total score [31], 2) 6MWD [12], 3) 4SC completion time, measured in seconds, and 4) 4SC velocity, measured in stairs/second.

The NSAA evaluates motor function in ambulatory DMD based on 17 activities scored by trained clinical staff as 0 (unable to perform independently), 1 (performs activity using a modified method but is able to complete independently), or 2 (able to perform independently without modification). The NSAA total score is the sum of these activity scores and ranges from 0 (worst function) to 34 (best function). Protocols for administration of the NSAA in each data source are summarized in **S4 Table**.

Assessments of 6MWD in the studied sources were based on modified American Thoracic Society (ATS) criteria and administered by trained assessors or clinical experts [12]. Training procedures are described for each data source in **S5 Table**. Participants who were unable to complete the six-minute walk test at the time of their assessment were assigned a 6MWD of zero meters.

For 4SC, both completion times and velocities were studied, as both of these metrics have been used in clinical trials. The velocity scale, calculated as 4 divided by the 4SC completion time and reported in units of stairs/second, addresses skewness in the distribution of completion times arising from very long completion times for patients with poor function. Patients

Therapeutics, Sarepta, Santhera, and Pfizer; has served as PI for GlaxoSmithKline, Prosensa, BioMarin Pharmaceutical, Italfarmaco, Roche, PTC, Pfizer , Sarepta, Santhera, Wave , NS and Eli Lilly. EM reports personal fees as consultant, PI or member on advisory board form BIOGEN S.R.L. outside the submitted work; EM reports personal fees consultant, PI or member on advisory board from ROCHE; EM reports personal fees consultant, PI or member on advisory board form AVEXIS outside the submitted work; EM reports personal fees consultant, PI or member on advisory board from SCHOLAR ROCK outside the submitted work; EM is part of an institution that receives funding from Biogen for a SMA disease registry (ISMAR). Erik H. Niks is a member of the European Reference Network for Rare Neuromuscular Diseases (ERN EURO-NMD). EN report grants from Duchenne Parent Project, ZonMW and AFM, consultancies for BioMarin and Summit, and worked as local investigator of clinical trials of BioMarin, GSK, Lilly, Santhera, Givinostat, and Roche outside the submitted work. E.H.N. reports ad hoc consultancies for WAVE, Santhera, Regenxbio, and PTC, and he worked as investigator of clinical trials of Italfarmaco, NS Pharma, Reveragen, Roche, WAVE, and Sarepta outside the submitted work. Brenda Wong has participated in advisory committee meetings for Prosensa and Biomarin and has received compensation for consultancy services for Gilead Sciences, Pfizer, GSK, RegenXBio, and PepGen. Krista Vandenborne has received grants from NIH National Institute of Arthritis and Musculoskeletal and Skin Diseases/ National Institute of Neurologic Disorders and Stroke, Parent Project Muscular Dystrophy, and the Muscular Dystrophy Association. She has also received funding from Italfarmaco SpA, Sarepta Therapeutics, Summit Therapeutics plc, Catabasis Pharmaceuticals, Pfizer Inc, Idera Pharmaceuticals, Bristol-Myers Squibb, and Eli Lilly through grant awards to the University of Florida. Volker Straub has participated in advisory boards for Audentes Therapeutics, Biogen, Exonics Therapeutics, Italfarmaco S.p.A., Roche, Sanofi Genzyme, Sarepta Therapeutics, Summit Therapeutics, UCB, and Wave Therapeutics. He has research collaborations with Ultragenyx and Sanofi Genzyme. Imelda JM de Groot has no disclosures. Cuixia Tian has participated as the site principal investigator for trials sponsored by PTC Therapeutics, Eli Lilly, GSK, Prosensa/Biomarin, Bristol Myers Squibb, Roche, Pfizer. Santhera, Sarepta, Fibrogen, Capricor, Pfizer, Avexis, and Catabasis. Adnan Manzur has no disclosures. Susan J. Ward co-founded and manages the collaborative Trajectory Analysis Project and has

who were not able to complete the 4SC due to loss of function were assigned 4SC velocities of zero. In scaling the 4SC to stairs/second, we represent the average velocity across all four stairs without assuming or implying that velocities are constant across all four stairs. Conduct and recording of the timed 4SC can vary across RWD/NHD and clinical trial settings, particularly for patients with poor function. In RWD sources, for example, assessors may not require a patient with very poor function to attempt the test, resulting in missing completion times without a recorded reason. In contrast, protocol-driven assessments in clinical trials yield more complete conduct and recording of the 4SC, and thus greater representation of longer completion times. To assess the impact of differences in data collection settings we conducted separate analyses, stratified by type of data source (clinical trial or RWD/NHD), while also truncating 4SC completion times at 12 seconds or 30 seconds across all data sources. Completion times greater than 12 seconds are rare in RWD/NHD sources; thresholds of 30 seconds are commonly applied in clinical trials.

In addition to the outcome measures listed above, the present study used the Functional Motor Scale (FMS) as an anchor for the estimation of MCIDs for worsening. The FMS, an 8-point physician grading of motor function [31], was only available from CCHMC data. Higher FMS scores correspond to greater functional impairment: a score of 1 indicates mild abnormalities in gait and ability to climb stairs without assistance; a score of 2 indicates more apparent gait abnormalities and requirement for a railing or support for stairs; a score of 3 indicates ability to walk and arise from a chair independently, but inability to negotiate stairs without help; a score of 4 indicates that independent walking is the primary means of mobility, but that a walker, braces or other means of support is necessary, along with inability to rise from a chair independently; scores of 5 or higher indicate that a wheelchair is the primary or necessary means of mobility.

## Statistical methods

Three different methods were applied to estimate meaningful differences in function, each corresponding to a different concept of meaningfulness: (1) estimation of MDCs, (2) estimation of anchor-based MCIDs and (3) a supportive distribution-based method using a one-half of the measure's standard deviation in the study population.

**MDCs.** Patients included in the estimation of MDCs were aged $\geq 4$ to $<18$ years and receiving steroids at the start of their follow-up. Separate study samples were drawn for each outcome measure, with patients required to have a minimal level of baseline motor function on the specified outcome: NSAA >12, 6MWD >75 meters or 4SC time <12 seconds. A patient's first visit meeting all of these criteria served as his index visit. Patients with fewer than two outcome assessments following the index assessment were excluded.

MDCs were estimated via longitudinal modeling of functional trajectories as described by Van der Elst et al. [32]. In particular, patient-specific functional trajectories over time were modeled with spline curves, based on both fixed and random effects of age and adjustment for data source in a mixed effects model. Best fitting models were selected based on the Akaike Information Criterion (AIC) to determine the appropriate level of flexibility in the splines. In this formulation, the fitted curve for each patient represents their true, underlying functional trajectory, such that deviations of the observed functional assessments above and below that curve represent measurement error and other transient factors influencing measured function. The distribution of these deviations was inspected for symmetry and summarized across all patients as the residual standard error (RSE) for the fitted model (**Fig 1**). MDC thresholds corresponding to 80% and 90% confidence in true change (improvement or worsening) were calculated based on RSEs as described in **S1 Text**. To assess the sensitivity of these estimates, analyses were stratified by type of data source, age and baseline function.

received funding from the membership of cTAP to facilitate this study. Laurent Servais is member of the SAB or has performed consultancy for Sarepta, Dynacure, Santhera, Avexis, Biogen, Cytokinetics and Roche, Audentes Therapeutics and Affinia Therapeutics; LS has given lectures and has served as a consultant for Roche, Biogen, Avexis, and Cytokinetics. LS is the project leader of the newborn screening in Southern Belgium funded by Avexis, Roche, and Biogen.]. This does not alter our adherence to PLOS ONE policies on sharing data and materials.

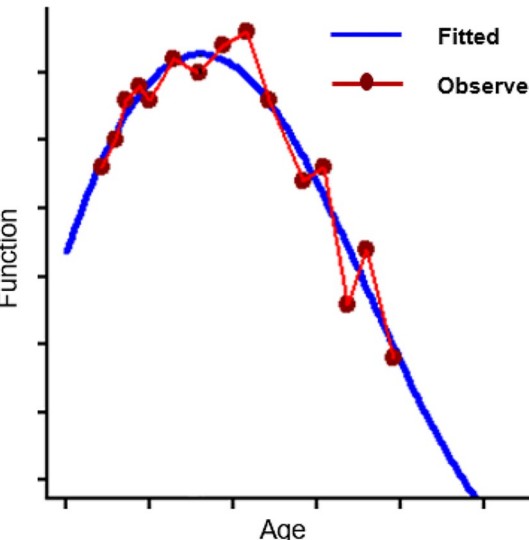

**Deviations from the individual's fitted curve represent measurement error and/or transient biological variation, not true progression.**

**Model-based estimates of variation around the fitted curve (residual standard error) provide an estimate of the standard error of measurement.**

**Fig 1. Illustration of an individual's fitted and observed functional trajectory, and estimation of the standard error of measurement.**

**Anchor-based MCIDs.**   Anchor-based MCIDs were estimated using the FMS score as an anchor within data from CCHMC. Included patients were aged ≥6 to <18 years, receiving steroids for at least 6 months, had a starting FMS score of 1 or 2, and functional assessments spanning intervals of approximately 48 weeks. Regression models were used to estimate the mean 48-week change in 4SC velocity or NSAA total score for patients with a 1-point worsening in FMS score (i.e., from 1 to 2, or from 2 to 3) versus those with no change in FMS over the 48-week period. Very few patients with 4SC and NSAA assessments over 48 weeks experienced a >1 point worsening. (Likewise, very few patients experienced an improvement in FMS (i.e., from 2 to 1) Given this small number of improving patients, we did not have sufficient data to estimate separate MCIDs for improvement. Generalized estimating equations were used to account for use of multiple non-overlapping intervals of follow-up from individual patients. An anchor-based MCID analysis was not conducted for the 6MWD because none of the studied data sources included the 6MWD alongside the FMS or other suitable anchors. Pearson correlations between changes in the FMS anchor and changes in the NSAA and 4SC outcomes were measured to assess the suitability of the anchor.

**Half standard deviation.**   As a reference point, half-SD estimates [33] for each measure were calculated across all patient visits in the pooled sample overall and stratified by age. The half SD is a commonly used, distribution-based approach recommended as supportive for estimation of meaningful change [34], though, unlike the MDC and anchor-based MCID, it is without specific grounding in clinical meaning or statistical reliability [33].

## Results

Data source descriptions, including characterization of steroid use, are provided in **S1 and S2 Tables**.

### MDCs

Sample sizes, mean ages, and follow-up times for the MDC analyses of NSAA total score, 6MWD and 4SC are summarized by data source in **S3 Table**.

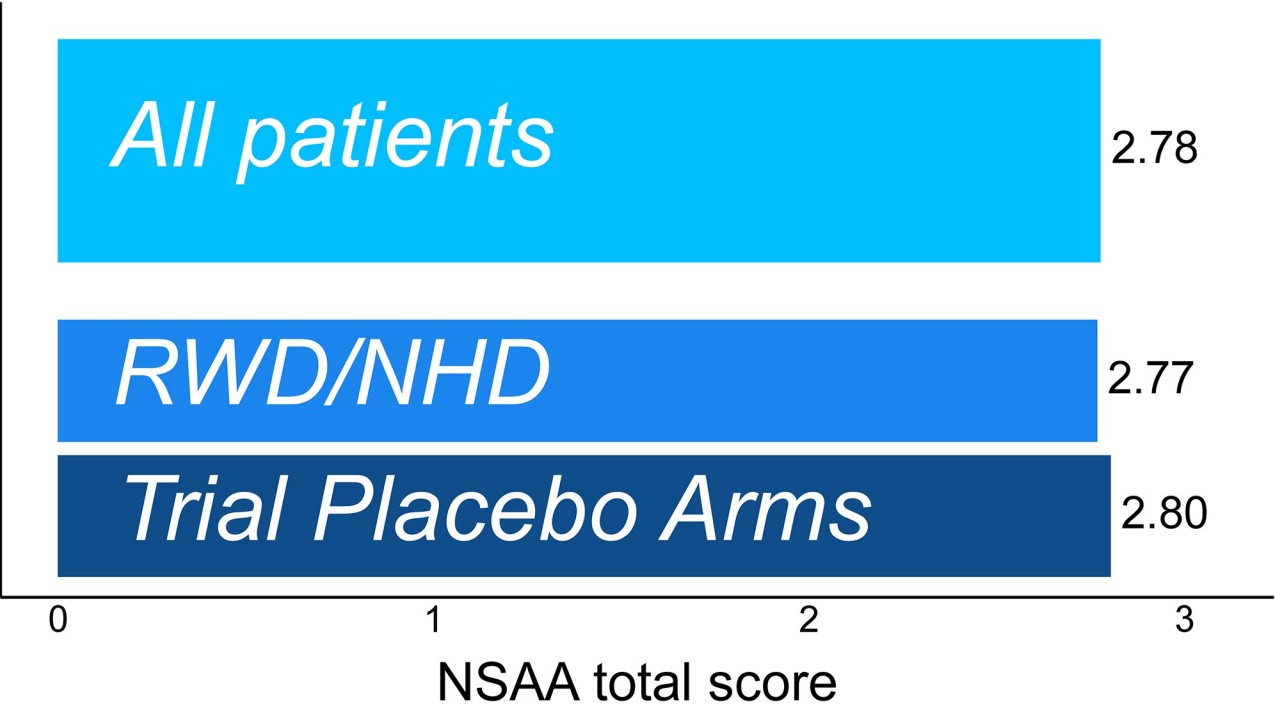

**Fig 2. Magnitude of change in NSAA total score required to have 80% confidence that true change has occurred.**

**NSAA total score.** A total of 5,917 assessments from 1,012 patients were studied (1,598/ 269 assessments/patients from clinical trials [median follow-up time: 11.1 months] and 4,319/ 743 assessments/patients from NHD studies [median follow-up time: 35.6 months]). The MDC threshold indicative of >80% confidence in true change in NSAA total score was 2.8 units in the pooled data from all sources, and almost identical in RWD/NHD and trial placebo arms (2.77 and 2.80 units, respectively) (**Fig 2**). MDC estimates in individual data sources ranged from 2.5 to 3.2 units (**S6 Table**). The pooled MDC for 90% confidence was 4.2 units.

**6MWD.** A total of 3,701 assessments from 625 patients were studied (2,135/350 assessments/patients from clinical trials [median follow-up time: 11.0 months] and 1,566/275 assessments/patients from NHD studies [median follow-up time: 29.6 months]). The MDC threshold for >80% confidence in true change in 6MWD was 36.3 meters in the pooled data from all sources, 39.1 meters in natural history studies and 34.2 meters in trial placebo arms (**Fig 3**). MDC estimates in individual data sources ranged from 30.5 to 41.6 meters (**S7 Table**). The pooled MDC for 90% confidence was 54.5 meters.

**4SC time and velocity.** A total of 6,402 assessments from 1,029 patients were studied (2,436/425 assessments/patients from clinical trials [median follow-up time: 11.0 months] and 3,966/604 assessments/patients from NHD studies [median follow-up time: 36.2 months]). When 4SC completion times were truncated at 12 seconds, the MDC threshold indicative of >80% confidence in true change in 4SC time was 1.30 seconds in the pooled data from all sources. Estimates were almost identical in RWD/NHD and trial placebo arms (1.30 and 1.31 units, respectively) (**Fig 4**). MDC estimates in individual data sources for 4SC time truncated at 12 seconds ranged from 1.05 to 1.90 units (**S8 Table**). When truncating completion times at 30 seconds, MDC estimates at 80% confidence were systematically higher, ranging from 1.9 to 5.0 seconds across data sources (**S9 Table**).

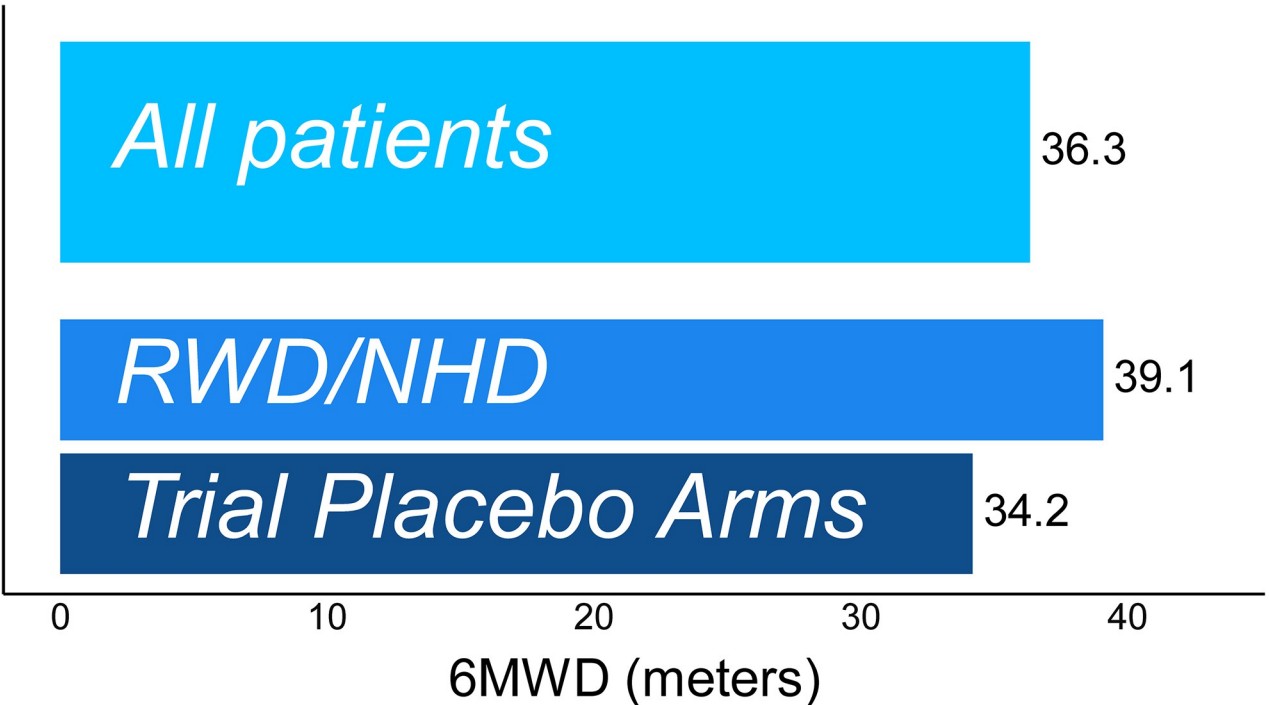

**Fig 3. Magnitude of change in 6MWD (meters) required to have 80% confidence that true change has occurred.**

When 4SC outcomes were studied on the velocity scale, MDCs were less sensitive to whether completion times were truncated at 12 seconds or at 30 seconds. The MDC thresholds for 4SC velocity with 80% confidence was 0.35 stairs/second when pooled across all data sources, regardless of truncation threshold. For truncation at 30 seconds, estimated MDCs with 80% confidence were 0.36 and 0.33 stairs/second, respectively, for NHD/RWD and clinical trials; source-specific estimates ranged from 0.27 to 0.44 stairs/second (**Fig 5**, **S8** and **S9 Tables**). The pooled MDC for 90% confidence was 0.53 stairs/second.

**Subgroups based on age and function.** As clinical trials in DMD often require enrolment by age and baseline function, MDC estimates were explored in subgroups based on these factors. For NSAA total scores, MDC estimates were similar across all three age groups (i.e., $\leq 7$, 7–12, and >12 years; ranging from 2.7 to 2.8 NSAA total score units) and were slightly higher for boys with lower vs. higher baseline function (**S6 Table**).

For 6MWD, the MDC estimates were higher in younger boys: 43.4 meters in boys aged $\leq 7$ years compared to 35.6 meters among boys aged 7–12 years or older than 12 years. The MDC was also higher for boys with lower baseline 6MWD (60 meters among those with baseline 6MWT between 75 and 200 meters) than for boys with higher baseline 6MWD performance (~34 meters in boys with baseline 6MWT above 200 meters) (**S7 Table**).

For 4SC completion times, MDC estimates were lower among younger boys, and higher among patients with worse baseline function (**S8** and **S9 Tables**). For 4SC velocity, MDC estimates were similar across age groups, and higher among patients with better baseline function (**S8** and **S9 Tables**).

### Anchor-based MCIDs

For the NSAA total score, 156 patients (306 48-week intervals) were studied. An average change of 2.2 units in the NSAA total score was associated with a 1-unit worsening in the

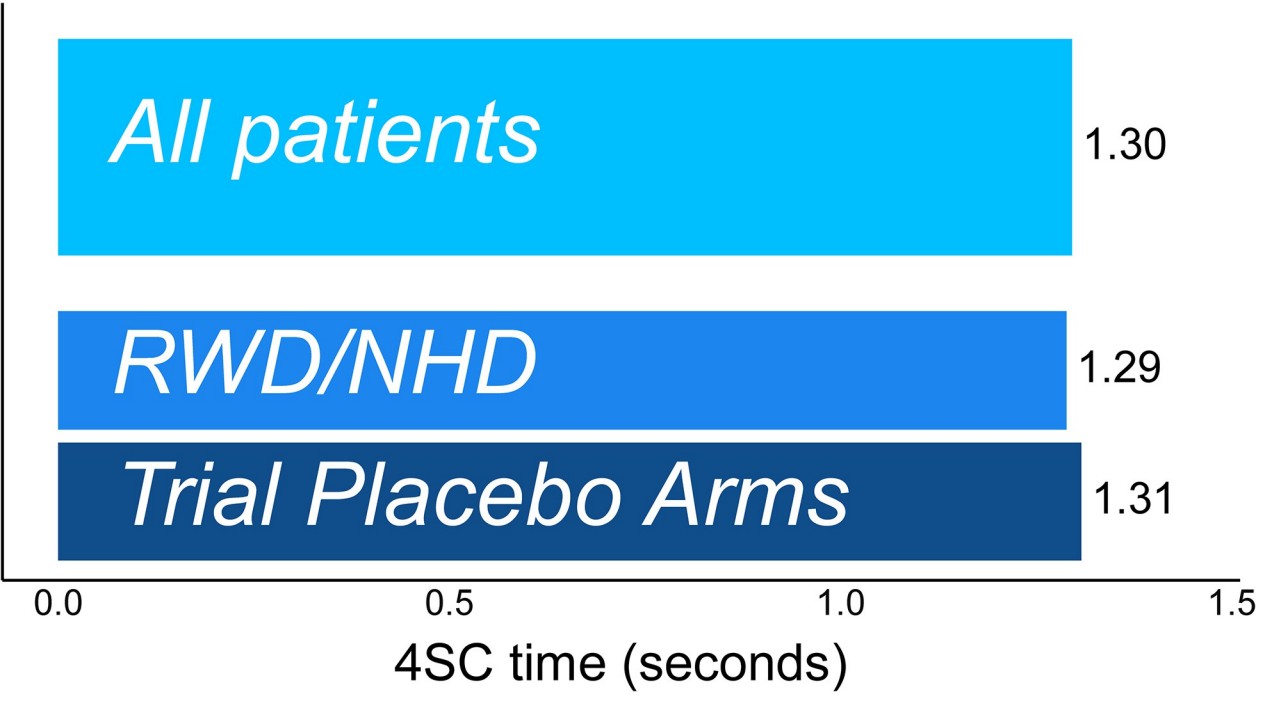

**Fig 4. Magnitude of change in 4SC time (seconds) required to have 80% confidence that true change has occurred.**

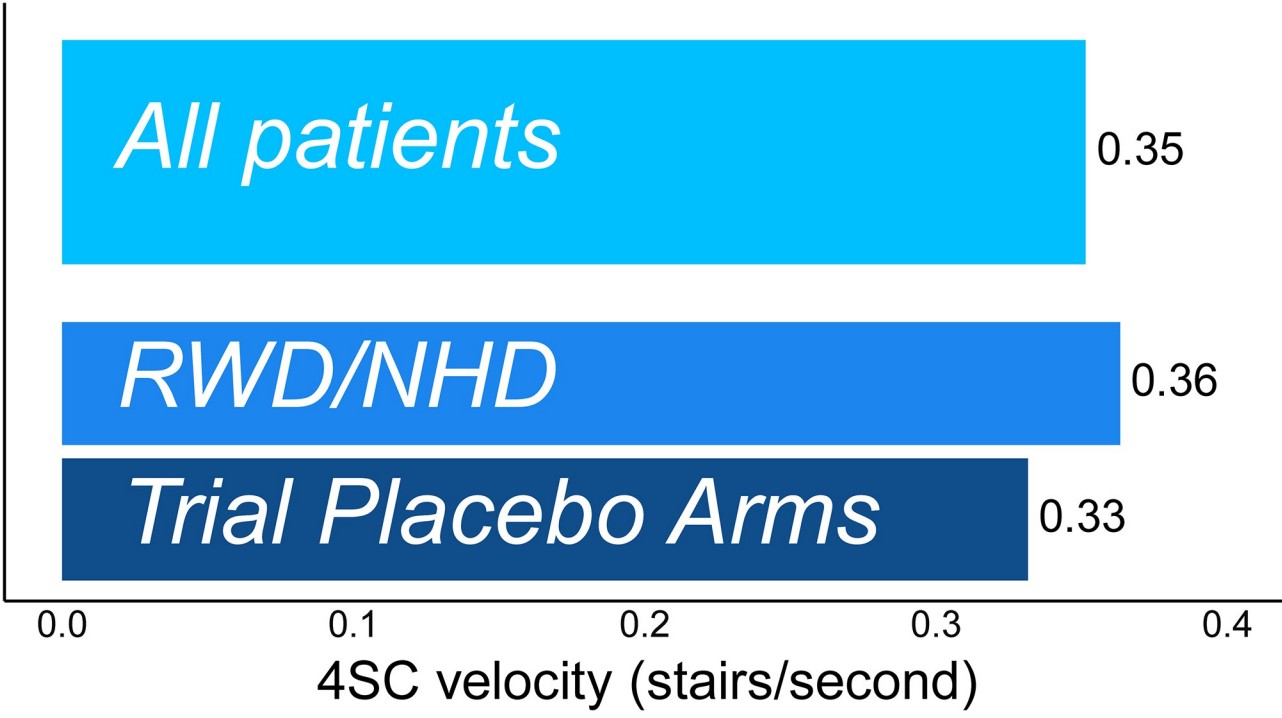

**Fig 5. Magnitude of change in 4SC velocity (stairs/second) required to have 80% confidence that true change has occurred.**

FMS. For the 4SC velocity, 182 patients, (376 intervals) were studied. An average change of 0.30 stairs/second in the 4SC was associated with a 1-unit worsening in FMS. Pearson correlations between changes in the FMS anchor and changes in NSAA and 4SC were small, at -0.24 and -0.20, respectively.

### Half-SD

Half-SD measures were 4.4 units for NSAA total score (ranging from 2.3 to 5.4 across 1-year age groups) and 79.4 meters for 6MWD (ranging from 34.4 to 90.9 across 1-year age groups). When 4SC completion times were truncated at 12 seconds, the half-SD was 1.9 seconds (ranging from 0.6 to 2.1 across 1-year age groups). When completion times were truncated at 30 seconds, the half-SD for 4SC was 4.8 seconds (ranging from 0.6 to 6.4 across 1-year age groups). Half-SD estimates for 4SC velocity were similar regardless of truncation times: 0.42 and 0.44 stairs/second, respectively, when truncating at 12 and 30 seconds, with estimates ranging from these levels to as low as 0.29 stairs/second across 1-year age groups.

## Discussion

The primary finding of this study was that if an individual patient with DMD worsened in measured function by at least 3 units of the NSAA total score, or 40 meters of 6MWD, or 0.35 stairs/second for the 4SC, then in each case one can be at least 80% confident that he had experienced disease progression. The corresponding thresholds for >90% confidence were 5 units for the NSAA, 55 meters for 6MWD and 0.55 stairs/second for the 4SC. The inverse also held: functional improvement exceeding these thresholds indicated, with the corresponding levels of confidence, that a patient had truly improved their function. These MDC thresholds slightly exceeded MCIDs for worsening estimated for NSAA total score (2.2 units) and 4SC velocity (0.3 stairs/second) from the present study and were in the range of MCIDs estimated for 6MWD from a prior study (6 to 46 meters) [23]. Individual patient changes in function exceeding these MDCs are likely to be clinically meaningful.

These MDC thresholds were highly consistent for the NSAA total score across the studied data sources and subpopulations. For the other measures, 6MWD and 4SC velocity, special attention should be paid to the baseline level of motor function when applying MDCs to specific populations, as the estimated MDCs were found to be more sensitive to this characteristic. All findings were based on steroid-treated boys and may not generalize to steroid-naïve boys.

### Consistency of MDC estimates across data sources

MDC estimates for the NSAA total score and 6MWD were similar across pooled clinical trial arms and pooled RWD/NHD sources. Differences among individual data sources were small relative to the size of the MDCs. This may be due to the broad consistency in assessment protocols and clinical evaluators for these two tests across clinical trials and the RWD/NHD sources included here. The importance of standardized evaluator training for the reliability of NSAA scores is well-documented [13]. The consistency of levels of variation in NSAA total scores in the present study, as reflected in estimated MDCs across multiple RWD/NHD and clinical trial settings, suggests that such training efforts have been effective across the included care centers. The fact that many centers in which RWD/NHD were collected were also clinical trial centers may have also contributed to the similarities in MDC estimates across data sources.

In contrast, MDC estimates for the 4SC completion time varied across data sources, most likely due to systematic differences in 4SC test conduct and recording. In particular, assessors in real-world clinical practice may not require a patient to attempt a 4SC test when very poor function or inability is anticipated, and the test becomes very burdensome for patients. Within

clinical trials, in contrast, protocol-driven consistency in assessment is more likely. Truncation of 4SC completion times at 12 or 30 seconds increased the consistency across data sources, though sensitivity to the specific truncation threshold remained. We also investigated MDCs for the 4SC velocity which, as the mathematical inverse of the completion time, is less sensitive to outlying high completion times. The MDCs for 4SC velocity were highly consistent across data sources and choice of truncation threshold. We conclude that 4SC velocities are more robust to differences in test conduct across care settings. Greater consistency in the conduct and recording of 4SC completion times–especially consistency in how long to allow an unassisted attempt and objective documentation of inability or the decision not to conduct a test–would help improve the consistency of 4SC completion times across RWD/NHD and clinical trial settings.

## Patient populations relevant for clinical trials

We investigated whether MDCs varied numerically across sub-populations defined by age and function, since clinical trials use various such criteria for enrollment. For NSAA, 6MWD, and 4SC velocity, the MDC estimates were generally similar across different baseline age groups, whereas for 4SC time, MDC estimates were higher in older age groups. Among patients starting from lower levels of function, slightly larger changes in NSAA, 6MWD and 4SC completion times were required to be confident that true change had occurred, suggesting that these assessments can have slightly more variability as function worsens. Non-assessment, and missing functional data, as boys approach loss of function could also contribute to this pattern. For 4SC velocity, the opposite pattern was seen, wherein MDC estimates were larger for patients starting with better function. This is consistent with the inverse relationship between 4SC completion time and velocity, and the greater sensitivity of very fast velocities to a given small difference in completion time. Another contributor could be variation in whether or not a boy with better function chooses to use the rails when ascending stairs as quickly as he can, which could be addressed with greater standardization of the 4SC test instructions. Overall we observed greater sensitivity of MDCs to baseline function than to baseline age, consistent with prior studies showing that baseline function explains more variability in functional change than baseline age in DMD [6,35].

## Other studies of meaningful change

Other studies have reported estimates of meaningful change for 6MWD and 4SC in DMD. A longitudinal study in 2013 [7] estimated the standard error of measurement (SEM) for 6MWD and 4SC time using pre-treatment data from 174 patients enrolled in the ataluren phase 2b trial. Estimated SEMs in that study were 28.5 meters for 6MWD and 2.1 seconds for 4SC time [7], which would yield MDC estimates at the 80% confidence level of 34.2 meters and 2.5 seconds. These single-trial estimates are well-within the ranges identified in the present study across multiple data sources, though slightly lower than our pooled MDC estimates. An important difference in methodology and data, however, is that the estimates of test-retest reliability in the 2013 study were based on screening and baseline assessments taken up to 6 weeks apart among those patients who met all screening criteria and returned for baseline. The estimates in the present study are based on longer periods of longitudinal follow-up, and are thus accounting for transient biological variation that might extend beyond a 6 week test-retest period. For this reason, when applying MDCs to identify changes in function over 1-year, or longer periods, we would recommend use of the thresholds estimated from the current study.

It should be noted that there has been no general consensus on how MDC is defined in general or in DMD. A recent study published in 2021 [36] has reported MDC estimates for timed

function tests in DMD using a different concept and definition of MDC: "the 12-month change in time function score divided by the standard deviation of each time function outcome at baseline," which can also be referred to as an "effect size" (e.g., [37,38]). Despite both the 2021 study and the present study having reported quantities labeled as "MDC," these studies are in fact measuring different concepts (i.e., not simply different approaches to studying the same concept). For precision and to avoid confusion we present an example. Suppose a population was composed of patients with both increasing and decreasing levels of function such that the average change in function was zero. The MDC estimates in the present study would likely remain unchanged whereas the MDC estimates from the approach applied to the other study would necessarily be zero. The interpretation of the MDC estimate in that study, in this case, would be that mean change in the studied population is negligible as a proportion of baseline variation. The interpretation of the present study's MDC estimate would be, as described above, the minimum level of change that needs to be observed for an individual patient to be confident he has experienced true change. Another recent study has calculated the standard error of measurement (SEM) for the NSAA total score as 2.9–3.5 points across ages 7–10 years [24] using intra-class correlations coefficients derived from a prior study [39].

Finally, previous studies have reported proxies for meaningful change or MCIDs based on the half-SD or 1/3 SD methods or test-retest data [24]. As has been found for many other metrics, we find that half-SDs are a reasonable and conservative proxy for MCIDs for broad populations with ambulatory DMD. However, especially for 6MWD and 4SC completion times, the half-SD can vary substantially across populations and can become overly conservative.

### Other studies of anchor-based MCIDs

MCID estimates for 6MWD in DMD have been previously reported as ranging from 5.6 to 45.9 meters, depending on the level of baseline function [23], in a study of 24 boys in which the PODCI transfer and basic mobility scale was used as an anchor. Another study has reported MCIDs for the 4SC and other timed function tests using the Vignos scale as an anchor and a different statistical methodology (predicting changes in the anchor using logistic regression) [36]. The first two score levels of the Vignos scale are very similar to score levels 1 and 2 of the FMS, which was used as an anchor in the present study. The MCID for 4SC was estimated in this prior study as 0.035 tasks per second, corresponding to 0.14 stairs/second in the units of the present study and representing the degree of change in 4SC velocity required to predict a 1-unit change in the Vignos scale. The degree of accuracy in that prediction was low (AUC < 0.7) [36], consistent with the weak correlation between 4SC and FMS observed in the present study. These MCID estimates for 4SC are not directly comparable given differences in data and methodologies, but provide confidence that changes exceeding MDCs are meaningful to patients. An anchor-based MCID for the NSAA total score, anchoring on the 6MWD, has been estimated at 3.5 units [24]. This study also conducted patient and caregiver interviews, which identified complete loss of function in a single NSAA item or deterioration of function in one to two items to be an important change.

### Applications to clinical trials and clinical practice

An application of MDCs to clinical trials is under separate investigation by our group: a functional progression endpoint defined as the time from baseline to worsening function that exceeds the MDC. For example, our estimated MDC of 2.8 units for NSAA total, may be considered a minimum threshold for a 'progressor' definition, such that any patient worsening beyond this threshold is considered to have a true worsening in function. Such endpoints may enable event-driven trials, which can reduce the duration of placebo exposure based on

individual rates of progression and overall treatment efficacy. Beyond clinical trials, MDCs can help clinicians distinguish progressive functional change from transient variation in measured function, complementing their overall clinical judgement about a patient's individual disease course and treatment recommendations.

When applying the current MCIDs to the interpretation of treatment effects in DMD clinical trials, the following should be kept in mind. First, the presence of the word "minimal" in MCID should not be over-interpreted. It is possible that differences in motor function smaller than all of these estimates are meaningful for individual patients or as between-group averages in DMD. The "minimal" in MCID means only that larger differences in function are assuredly meaningful; the meaningfulness of smaller difference is not ruled out. Considering that the anchors used for MCID estimation in DMD have been either non-specific to DMD (e.g., PODCI domains), specific functional tests (e.g., 6MWD) or broad functional categories (e.g., FMS in the present study) we would expect that smaller changes in motor function than the reported MCIDs are very likely meaningful to patients and caregivers. Indeed, recent interviews with DMD patients and caregivers have demonstrated the importance of loss of function on a single NSAA item or deterioration of function in one to two items [24]. In fact the FMS, as used in the present study—where it was available in only one data source—should be rated as a low quality anchor for MCIDs in Duchenne [40]: FMS is not assessed directly by patients, changes in FMS have low correlations with changes in NSAA and 4SC, and the increments of FMS represent too large a difference in motor function. We think this makes the FMS a highly conservative anchor that most likely over-estimates MCIDs–i.e., smaller-than-MCID changes in NSAA and 4SC are likely meaningful to patients. More sensitive quantification of the clinical meaning associated with motor function changes in DMD is sorely needed to help decision-makers interpret the effects of novel therapeutics studied in clinical trials.

## Conclusions

This study was conducted by cTAP, a pre-competitive consortium of academic collaborators, patient foundations and drug developer, and included more than 1,000 patients with DMD across more than 30 clinical care centers for neuromuscular disease from multiple institutions, registries and countries. Comparisons between NH/RWD and placebo arm data, and the ability to study sub-groups of patients, was only possible by analyzing data from a large number of patients, emphasizing the importance of data sharing and collaboration for DMD research. Overall, given the relative consistency of the findings across these multiple different data sources, the thresholds reported can inform the interpretation of functional changes in DMD clinical practice and clinical trials.

## Supporting information

**S1 Text. Supplementary methods.**
(DOCX)

**S1 Table. Description of clinical trial data sources.**
(DOCX)

**S2 Table. Description of real-world and natural history data sources.**
(DOCX)

**S3 Table. Sample size, median follow-up and mean age at first visit in MDC analyses.**
(DOCX)

**S4 Table. North Star Ambulatory Assessment details.**
(DOCX)

**S5 Table. 6MWD assessment details.**
(DOCX)

**S6 Table. Magnitude of change in NSAA total required to have 80% or 90% confidence that true change has occurred, among all patients, by data source, and by subgroups of function and age.**
(DOCX)

**S7 Table. Magnitude of change in 6MWD (meters) required to have 80% or 90% confidence that true change has occurred, among all patients, by data source, and by subgroups of function and age.**
(DOCX)

**S8 Table. Magnitude of change in 4SC time (seconds, with completion times truncated at 12 seconds) and 4SC velocity (stairs/second) required to have 80% or 90% confidence that true change has occurred, among all patients, by data source, and by subgroups of function and age.**
(DOCX)

**S9 Table. Magnitude of change in 4SC time (seconds, with completion times truncated at 30 seconds) and 4SC velocity (stairs/second) required to have 80% or 90% confidence that true change has occurred, among all patients, by data source, and by subgroups of function and age.**
(DOCX)

## Acknowledgments

The authors are grateful to patients for participating in the clinical assessments and agreeing to make their data available for research.

The authors would like to thank each member of the following author groups:

**PRO-DMD-01:** Nathalie Goemans (lead author; email: nathalie.goemans@uzleuven.be; Child Neurology, University Hospitals Leuven, Leuven, Belgium); Nicolas Deconinck (Department of Pediatric Neurology, Hôpital Universitaire des Enfants Reine Fabiola, Université Libre de Bruxelles, Brussels, Belgium); Mar Tulinius (Queen Silvia Children's Hospital, Gothenburg, Sweden); Kevin Flanigan (Nationwide Children's Center for Gene Therapy, Columbus, Ohio, USA); Erik Henricson (Department of Physical Medicine and Rehabilitation, School of Medicine, University of California, Davis, California, USA); Maria Bernadete Dutra de Resende (Department of Neurology, University of São Paulo, Brazil); Gian Luca Vita (Nemo Sud Clinical Centre, University Hospital "G. Martino", Messina, Italy); Ulrike Schara and JanBerd Kirschner (Universitätsklinikum Freiburg, Freiburg, Germany); Haluk Topaloglu (Hacettepe University Medical Faculty/Pediatric Neurology, Sıhhiye-Ankara, Turkey); Soledad Monges (Department of Neurology, Hospital J P Garrahan, Argentina); and Claude Cances (Unité de Neuropédiatrie, Hôpital des enfants, CHU Toulouse, Toulouse, France).

**Association Française contre les Myopathies:** Francesco Muntoni (lead author; email: f.muntoni@ucl.ac.uk; Dubowitz Neuromuscular Centre, NIHR Great Ormond Street Hospital Biomedical Research Centre, Great Ormond Street Institute of Child Health, University College London, & Great Ormond Street Hospital Trust, London, United Kingdom); Joana Domingos (deceased) and Valeria Ricotti (Dubowitz Neuromuscular Centre, University College

London Great Ormond Street Institute of Child Health and Great Ormond Street Hospital, London, United Kingdom); Victoria Selby, Amy Wolfe, Lianne Abbott, Evelin Milev, Efthymia Panagiotopoulou, Mario Iodice, and Maria Ash (Dubowitz Neuromuscular Centre, Great Ormond Street Hospital for Children NHS Trust, London, United Kingdom); Laurent Servais (MDUK Oxford Neuromuscular Center, Department of Paediatrics, University of Oxford, United Kingdom, and Neuromuscular Center of Liège, Division of Paediatrics, CHU and University of Liège, Belgium); Professor Thomas Voit ([former] Groupe Hospitalier Pitié Salpêtrière, Institut de Myologie, Paris, France; [current] Dubowitz Neuromuscular Centre, University College London Great Ormond Street Institute of Child Health, NIHR Great Ormond Street Hospital Biomedical Research Centre, London, United Kingdom); Valérie Decostre, Stéphanie Gilabert, and Jean-Yves Hogrel (Groupe Hospitalier Pitié Salpêtrière, Institut de Myologie, Paris, France); Volker Straub, Alexander Murphy, and Anna Mayhew (John Walton Muscular Dystrophy Research Centre, Newcastle University, Newcastle, United Kingdom); Menno Van der Holst (Department of Orthopaedics, Rehabilitation and Physiotherapy, Leiden University Medical Centre, Leiden, The Netherlands); Erik H. Niks, Yvonne D. Krom, and Marjolein J. van Heur-Neuman (Department of Neurology, Leiden University Medical Centre, Leiden, The Netherlands); Imelda JM de Groot, Merel Jansen, Maaike Pelsma, and Marian Bobbert (Department of Rehabilitation, Donders Centre of Neuroscience, Radboud university medical center, Nijmegen, The Netherlands); Johannes J.G.M. Verschuuren (Leiden University Medical Centre, Leiden, The Netherlands).

**UK NorthStar Clinical Network sites:** Francesco Muntoni (lead author; email: f.muntoni@ucl.ac.uk; Dubowitz Neuromuscular Centre, NIHR Great Ormond Street Hospital Biomedical Research Centre, Great Ormond Street Institute of Child Health, University College London, & Great Ormond Street Hospital Trust, London, United Kingdom); Adnan Manzur (lead author; email: adnan.manzur@gosh.nhs.uk; Dubowitz Neuromuscular Centre, NIHR Great Ormond Street Hospital Biomedical Research Centre, Great Ormond Street Institute of Child Health, University College London, & Great Ormond Street Hospital Trust, London, United Kingdom); Stephanie Robb, Rosaline Quinlivan, Anna Sarkozy, Pinki Munot, Giovanni Baranello, Mariacristina Scoto, Marion Main, Lianne Abbott, Hinal Patel, Salma Samsuddin, and Vandana Ayyar Gupta (Dubowitz Neuromuscular Centre, Great Ormond Street Hospital for Children NHS Trust, London, United Kingdom); Kate Bushby, Chiara Bertolli, Anna Mayhew, Robert Muni-Lofra, Meredith James, Dionne Moat, and Jassie Sodhi (Institute of Human Genetics, John Walton Muscular Dystrophy Research Centre, Newcastle); Helen Roper, Deepak Parasuraman, Heather McMurchie, and Rosanna Rabb (University Hospitals Birmingham NHS Foundation Trust); Karen Pysden and Lindsey Pallant (Yorkshire Regional Muscle Clinic, Leeds General Infirmary); G Peachey, Rajesh Madhu, and Alison Shillington (Alder Hey Childrens NHS Foundation Trust, Liverpool); Heinz Jungbluth, Jennie Sheehan, and R Spahr (Evelina London Childrens Hospital, Guy's and St Thomas' NHS Foundation Trust); E. Bateman and C. Cammiss (Royal Manchester Childrens Hospital, Manchester); L. Groves and Nicholas Emery (The Muscle Clinic, Robert Jones and Agnes Hunt Orthopaedic Hospital NHS Foundation Trust, Oswestry); P. Baxter, N. Goulborne, M. Senior, and E. Scott (Sheffield Childrens Hospital NHS Foundation Trust, Sheffield); L. Hartley and Bethan Parsons (Cardiff and Vale University Health Board); Faye Mason, L. Jenkins, and B. Toms (Bristol Royal Hospital for Children, University Hospitals Bristol NHS Foundation Trust, Bristol); Claire Frimpong-Ansah (University Hospitals Plymouth NHS Trust, Plymouth); Heather Jarvis (Plymouth Child Development Centre,Plymouth); J. Dalgleish and A. Keddie (Armistead Child Development Centre, Kings Cross Hospital, Dundee); Marina Di Marco and J. Dunne (Royal Hospital for Children, NHS Greater Glasgow and Clyde, Glasgow); A. Miah (Nottingham University Hospitals, Nottingham); Andrea Selley (Preston Royal Hospital, Lancashire

Teaching Hospitals NHS Foundation Trust); Michelle Geary and Jenni Palmer (Southampton Childrens Hospital, University Hospital Southampton NHS Foundation Trust, Southampton); Kate Greenfield (Abertawe Bro Morgannwg University Health Board, Swansea); S. MacAuley (Royal Belfast Hospital for Sick Children, Belfast); H. Robbins and M. Iqbal (Leicester Royal Infirmary, Leicester); Catherine Ward and Jacqui Taylor (Addenbrookes Hospital, Cambridge University Hospitals NHS Foundation Trust, Cambridge); A. OHara and Jane Tewnion (Royal Aberdeen Children's Hospital, Aberdeen); Saleel Chandratre, Sithara Ramdas, M. White, and Hayley Ramjattan (Oxford University Hospitals NHS Foundation Trust, Oxford); and J. Yirrel (Royal Hospital for Sick Children, Edinburgh).

**Imaging DMD:** Krista Vandenborne (lead author; email: kvandenb@phhp.ufl.edu; Department of Physical Therapy, University of Florida, Gainesville, Florida, USA); Harneet Arora, Rebecca J Willcocks, Donovan J Lott, Claudia R Senesac, William T Triplett, Barry J Byrne, Glenn A Walte, and HL Sweeney (University of Florida, Gainesville, Florida, USA); Ann T Harrington and Gihan I Tennekoon (The Children's Hospital of Philadelphia, Philadelphia, Pennsylvania, USA); Kirsten L Zilke, Erika L Finanger, and Barry S Russman (Oregon Health & Science University, Portland, Oregon, USA); Michael J Daniels and Dandan Xu (The University of Texas at Austin, Austin, Texas, USA); and Richard S Finkel (Nemours Children's Hospital, Orlando, Florida).

**cTAP:** The authors thank members of the cTAP Joint Steering Committee for their contributions to the conceptualization of the study, interpretation of the results, and review of the article, particularly Susan J. Ward PhD (lead author; email: susanjward@ctap-duchenne.org; cTAP); Paolo Bettica MD, PhD (Italfarmaco SpA) and Madeleine Billeter, MD PhD (Roche).

The authors would also like to acknowledge investigators and staff of the following RWD/ NHD sources included in this study:

**Universitaire Ziekenhuizen Leuven:** Marleen Van den Hauwe (Department of Child Neurology, University Hospitals Leuven, Leuven, Belgium).

**CCHMC:** Irina Rybalsky, K. Courtney Shellenbarger, Ann E. McCormick, Michelle N. McGuire, Kelly Bonarrigo, Amanda E. Fowler, Mike Kiefer, Jean Bange, and Shengyong Hu (Cincinnati Children's Hospital Medical Center, Cincinnati, Ohio, USA).

Flora Chik, Jingyi Chen and Pang Shen, employees of Analysis Group, Inc., assisted with medical writing and preparation of figures for this article.

## Author Contributions

**Conceptualization:** Francesco Muntoni, James Signorovitch, Susan J. Ward.

**Data curation:** James Signorovitch, Gautam Sajeev, Nicolae Done, Zhiwen Yao, Ibrahima Dieye, Henry Lane.

**Formal analysis:** James Signorovitch, Gautam Sajeev, Nicolae Done, Zhiwen Yao, Ibrahima Dieye, Henry Lane.

**Funding acquisition:** James Signorovitch, Susan J. Ward.

**Investigation:** Francesco Muntoni, Nathalie Goemans, Craig McDonald, Eugenio Mercuri, Erik H. Niks, Brenda Wong, Krista Vandenborne, Volker Straub, Imelda J. M. de Groot, Cuixia Tian, Adnan Manzur, Laurent Servais.

**Methodology:** James Signorovitch, Gautam Sajeev, Nicolae Done.

**Project administration:** James Signorovitch, Gautam Sajeev, Nicolae Done, Susan J. Ward.

**Resources:** Francesco Muntoni, Nathalie Goemans, Craig McDonald, Eugenio Mercuri, Erik H. Niks, Brenda Wong, Krista Vandenborne, Volker Straub, Imelda J. M. de Groot, Cuixia Tian, Adnan Manzur, Laurent Servais.

**Software:** James Signorovitch, Gautam Sajeev, Nicolae Done, Zhiwen Yao, Ibrahima Dieye, Henry Lane.

**Supervision:** James Signorovitch, Gautam Sajeev, Nicolae Done, Susan J. Ward.

**Validation:** James Signorovitch, Gautam Sajeev, Nicolae Done, Zhiwen Yao, Ibrahima Dieye, Henry Lane.

**Visualization:** James Signorovitch, Gautam Sajeev, Nicolae Done, Zhiwen Yao, Ibrahima Dieye, Henry Lane.

**Writing – original draft:** Francesco Muntoni, James Signorovitch, Gautam Sajeev.

**Writing – review & editing:** Francesco Muntoni, James Signorovitch, Gautam Sajeev, Nicolae Done, Zhiwen Yao, Nathalie Goemans, Craig McDonald, Eugenio Mercuri, Erik H. Niks, Brenda Wong, Krista Vandenborne, Volker Straub, Imelda J. M. de Groot, Cuixia Tian, Adnan Manzur, Ibrahima Dieye, Henry Lane, Susan J. Ward, Laurent Servais.

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
