## [Decision Letter · Decision Letter 0]

14 Dec 2023

PONE-D-23-24837Meaningful changes in motor function in Duchenne muscular dystrophy (DMD): a multi-center studyPLOS ONE

Dear Dr. Signorovitch,

Thank you for submitting your manuscript to PLOS ONE. After careful consideration, we feel that it has merit but does not fully meet PLOS ONE’s publication criteria as it currently stands. Therefore, we invite you to submit a revised version of the manuscript that addresses the points raised during the review process.

We look forward to receiving your revised manuscript.

Kind regards,

Jianhong Zhou

Staff Editor

PLOS ONE

[We have read the journal's policy and the authors of this manuscript have the following competing interests: 

Francesco Muntoni is a member of the Rare Disease Scientific Advisory Group for Pfizer and of Dyne Therapeutics SAB, and has participated to SAB meetings for PTC, Sarepta, Santhera, Wave Therapeutics. UCL and Great Ormond Street Hospital are recipient of grants from Pfizer, Italfarmaco, Wave, Santhera, Sarepta regarding clinical trials.

James Signorovitch co-founded the collaborative Trajectory Analysis Project (cTAP) and is an employee of Analysis Group, Inc., a consulting firm that received funding from the membership of cTAP to conduct this study.

Gautam Sajeev, Nicolae Done, and Henry Lane are employees of Analysis Group, Inc., a consulting firm that received funding from the membership of cTAP to conduct this study. Ibrahima Dieye and Zhiwen Yao were employees of Analysis Group, Inc. at the time this study was conducted. 

Nathalie Goemans has served on clinical steering committees and/or as a consultant and received compensation from Eli Lilly, Italfarmaco, PTC Therapeutics, and BioMarin Pharmaceutical; has served as site investigator for GlaxoSmithKline, Prosensa, BioMarin Pharmaceutical, Italfarmaco, Roche, and Eli Lilly.

Craig McDonald has served as a consultant for PTC Therapeutics, BioMarin Pharmaceutical, Sarepta Therapeutics, Eli Lilly, Pfizer Inc, Santhera Pharmaceuticals, Cardero Therapeutics, Inc, Catabasis Pharmaceuticals, Capricor Therapeutics, Astellas Pharma (Mitobridge), and FibroGen, Inc; serves on external advisory boards related to DMD for PTC Therapeutics, Sarepta Therapeutics, Santhera Pharmaceuticals, and Capricor Therapeutics; and reports grants from US Department of Education/National Institute on Disability and Rehabilitation Research, the National Institute on Disability, Independent Living, and Rehabilitation Research, US NIH/National Institute of Arthritis and Musculoskeletal and Skin Diseases, NIH/National Institute of Neurologic Disorders and Stroke, US Department of Defense, and Parent Project Muscular Dystrophy US.

Eugenio Mercuri has served on clinical steering committees and/or as a consultant for Eli Lilly, Italfarmaco, PTC Therapeutics, Sarepta, Santhera, and Pfizer; has served as PI for GlaxoSmithKline, Prosensa, BioMarin Pharmaceutical, Italfarmaco, Roche, PTC, Pfizer, Sarepta, Santhera, Wave, NS and Eli Lilly. EM reports personal fees as consultant, PI or member on advisory board form BIOGEN S.R.L. outside the submitted work; EM reports personal fees consultant, PI or member on advisory board from ROCHE; EM reports personal fees consultant, PI or member on advisory board form AVEXIS outside the submitted work; EM reports personal fees consultant, PI or member on advisory board from SCHOLAR ROCK outside the submitted work; EM is part of an institution that receives funding from Biogen for a SMA disease registry (ISMAR).

Erik H. Niks is a member of the European Reference Network for Rare Neuromuscular Diseases (ERN EURO‐NMD). EN report grants from Duchenne Parent Project, ZonMW and AFM, consultancies for BioMarin and Summit, and worked as local investigator of clinical trials of BioMarin, GSK, Lilly, Santhera, Givinostat, and Roche outside the submitted work. E.H.N. reports ad hoc consultancies for WAVE, Santhera, Regenxbio, and PTC, and he worked as investigator of clinical trials of Italfarmaco, NS Pharma, Reveragen, Roche, WAVE, and Sarepta outside the submitted work.

Brenda Wong has participated in advisory committee meetings for Prosensa and Biomarin and has received compensation for consultancy services for Gilead Sciences, Pfizer, GSK, RegenXBio, and PepGen.

Krista Vandenborne has received grants from NIH National Institute of Arthritis and Musculoskeletal and Skin Diseases/National Institute of Neurologic Disorders and Stroke, Parent Project Muscular Dystrophy, and the Muscular Dystrophy Association. She has also received funding from ltalfarmaco SpA, Sarepta Therapeutics, Summit Therapeutics plc, Catabasis Pharmaceuticals, Pfizer Inc, ldera Pharmaceuticals, Bristol-Myers Squibb, and Eli Lilly through grant awards to the University of Florida.

Volker Straub has participated in advisory boards for Audentes Therapeutics, Biogen, Exonics Therapeutics, Italfarmaco S.p.A., Roche, Sanofi Genzyme, Sarepta Therapeutics, Summit Therapeutics, UCB, and Wave Therapeutics. He has research collaborations with Ultragenyx and Sanofi Genzyme.

Imelda JM de Groot has no disclosures.

Cuixia Tian has participated as the site principal investigator for trials sponsored by PTC Therapeutics, Eli Lilly, GSK, Prosensa/Biomarin, Bristol Myers Squibb, Roche, Pfizer. Santhera, Sarepta, Fibrogen, Capricor, Pfizer, Avexis, and Catabasis.

Adnan Manzur has no disclosures.

Susan J. Ward co-founded and manages the collaborative Trajectory Analysis Project and has received funding from the membership of cTAP to facilitate this study.

Laurent Servais is member of the SAB or has performed consultancy for Sarepta, Dynacure, Santhera, Avexis, Biogen, Cytokinetics and Roche, Audentes Therapeutics and Affinia Therapeutics; LS has given lectures and has served as a consultant for Roche, Biogen, Avexis, and Cytokinetics. LS is the project leader of the newborn screening in Southern Belgium funded by Avexis, Roche, and Biogen.]. 

5. One of the noted authors is a group [Association Française contre les Myopathies, The UK NorthStar Clinical Network, ImagingDMD investigators, cTAP]. In addition to naming the author group, please list the individual authors and affiliations within this group in the acknowledgments section of your manuscript. Please also indicate clearly a lead author for this group along with a contact email address.

Reviewers' comments:

Reviewer's Responses to Questions

**Comments to the Author**

1. Is the manuscript technically sound, and do the data support the conclusions?

Reviewer #1: Yes

Reviewer #2: Yes

Reviewer #3: Partly

Reviewer #4: Yes

Reviewer #5: Yes

2. Has the statistical analysis been performed appropriately and rigorously? 

Reviewer #1: Yes

Reviewer #2: Yes

Reviewer #3: Yes

Reviewer #4: Yes

Reviewer #5: Yes

3. Have the authors made all data underlying the findings in their manuscript fully available?

Reviewer #1: Yes

Reviewer #2: Yes

Reviewer #3: Yes

Reviewer #4: Yes

Reviewer #5: Yes

4. Is the manuscript presented in an intelligible fashion and written in standard English?

Reviewer #1: Yes

Reviewer #2: Yes

Reviewer #3: Yes

Reviewer #4: Yes

Reviewer #5: Yes

5. Review Comments to the Author

Reviewer #1: The study was conducted by the collaborative Trajectory Analysis Project (cTAP), a precompetitive coalition of academic clinicians, drug developers, and patient foundations formed in 2015 to overcome the challenges of high variation in clinical trials in DMD. They sought to determine:

1. Estimated minimal detectable change (MDC) for selected motor function measures in ambulatory DMD i.e., the minimal degree of measured change needed to be confident that true underlying change has occurred rather than transient variation or measurement error

2. Minimal clinically important differences (MCIDs) i.e., differences in function perceived as beneficial to patients

They selected outcome measures commonly used in clinical trials and clinical practice 6MWD, NSAA and 4SC and evaluated data collected across multiple clinical sites, networks and clinical trial placebo arms.

Main findings include the estimated MDCs and MCIDs for these outcome measures, the impact of baseline functional ability, and that there is generally very good consistency in the way these outcome measures are assessed and scored across sites with trained investigators both in the real world and as part of clinical trials.

This is an impressive analysis of a large set of available data. It is very nice to see placebo arm data from clinical trials being made available and used alongside clinical centre or natural history data. It provides assurance that with trained staff excellent consistency can be achieved in assessments performed at multiple sites across the world. It provides important benchmarks for future clinical trials.

I have no specific critique.

Reviewer #2: Muntoni et al. determined minimal detectable changes and minimal clinically important differences for motor function tests (North Star Ambulatory Assessment, 4-stair climb and 6-minute walk distance) in ambulant Duchenne muscular dystrophy patients using available cohort data from clinical trial arms, real-world data and natural history studies, that represent up to thousand patients and several thousand assessments. Results were sub-analyzed in respect to different cohort data, as well as age and state of disease progression at baseline. Results are thus representative for the overall DMD population. NSAA, 4-SC and 6MWD are widely used tests to determine motor function in DMD, and routinely used by many care centers across the world for the follow-up of patients, as well as being outcome measures in many clinical trials. Therefore, herein estimated MDC and MCID, as being applicable to individual patients, will improve patient follow-up and therapeutic decisions, as well as help to judge treatment benefits. The importance of the herein presented data cannot be overstated. I’m certain that herein presented data will be extremely well perceived by the community. From a personal view, herein calculated MDS and MCID match my clinical experience, what I want to say is that I’m not surprised by the values that make entirely sense, and I certainly will feel very comforted in future when using these values for e.g. judging disease progression.

I would like to congratulate authors to this important work, and I should say that it was a great honor that I could, as reviewer, be one of the first readers. The manuscript is written perfectly throughout. Data analysis, as far as I can judge (as lacking expertise in statistics), is carefully performed. The introduction is informative and the discussion is critical. I have no other general or specific comments or requests to the authors. Bravo!

Reviewer #3: Muntoni et al. conducted a study to examine the minimal detectable change (MDC) for NSAA, 4-SC, and 6MWD in ambulatory individuals with DMD. Overall, the study is well-designed with high methodological quality, and the manuscript is well-written and structured.

I only have a few comments. Please see below.

(1) In addition to MDC, the study also assessed anchor-based MCIDs for the motor function measures mentioned above. The importance of the MCID has been highlighted by clinicians, regulatory authorities, patients/parents, and their advocacy groups. Linking changes in an outcome measure to clinical meaningfulness for patients is of great significance to regulatory agencies involved in drug approval processes. However, the findings regarding MCIDs are not very reliable and meaningful in this study. I believe removing relevant part of MCIDs will not do any harm to the manuscript. Firstly, MCIDs were not one of the main objectives of this study. Secondly, there was limited data source available for analyzing MCIDs. Thirdly, as stated by the authors, functional motor scale (FMS) used in this study is not an appropriate anchor for determining MCIDs in DMD patients: FMS is not self-assessed by patients themselves, changes in FMS have weak correlations with changes in NSAA and 4SC, and increments in FMS represent too large of a difference in motor function.

(2) Page 9, line 193

Reference #4 was not relevant to the FMS. I haven't used it previously. I've looked for references on the psychometrics of FMS, but couldn't find any. Can you please provide relevant references regarding the FMS?

Reviewer #4: This is a clinically meaningful research result that takes into account the use of RWD, which will become increasingly important in the future, and deserves to be adopted. There is nothing in particular to comment on.

Reviewer #5: This study tried to estimate the minimal detectable change (MDC) threshold > 80% confidence for selected motor function measures ( NSAA, 6MWT, 4SC) in ambulatory DMD, and the anchored based minimal clinically important differences (MCIDs) of NSAA or 4SC with 1 point worsening in FMS scores. The estimation is performed from the collected data of more than 1000 DMD ambulatory steroid treated patients , participating in either clinical trials placebo arms and real world clinical practice setting. It was found in this study that if an individual patient with DMD worsened in measured function by at least 3 units of the NSAA total score, or 40 meters of 6MWD,or 0.35 stairs/second for the 4SC, then in each case one can be at least 80% confident that he had experienced disease progression. The findings are useful for reference and the manuscript is well written.

After reading through the manuscript I have the following questions.

It was also mentioned that the study was trying to understand the meaningfulness of changes of these motor scales. It has been mentioned that such meaningfulness of changes can be estimated by following:

1. The minimal detectable change (MDC) threshold >80 % confidence or > 90% confidence for the selected motor function measures as a statistical estimated data as observed from a large number of DMD patients from clinical trials (placebo arms) and NH/RWD.

2. The meaningful changes to the families as in the changes in certain item in a certain motor measure.

3. The meaningful changes to the clinicians as using the FMS scores

While direct comparison of these different defined meaningful changes is not logical, what is the relationship of these different meaningful changes, and how these different meaningful dimensions supplementing each other worth further elaboration?

Can the author also explain why the MDC was higher for boys with lower baseline 6MWD (60 meters among those

with baseline 6MWT between 75 and 200 meters) than for boys with higher baseline 6MWD performance (~34 meters in boys with baseline 6MWT above 200 meters)?

In the abstract it was also mentioned the identified thresholds can be used to inform endpoint definitions, as inputs into power calculations, or as benchmarks for monitoring individual and group-level changes in motor function in ambulatory DMD. Does the authors mean primary outcome definition and input into power calculation of primary end point for clinical trial, or otherwise ? What are the group-level changes for benchmarking? As the finding in the specific group level changes found in this study has not been mentioned in the abstract. Can the author also included the specific group level benchmarking findings in the result part ?

For ambulatory steroid treated DMD patients, most will loss ambulation before or at 16 years old. Yet the current age range is up to <18. What is the percentage of DMD patients in the current collective cohort is between 16-18 age?

From what I understand that the current study are on DMD patients in the placebo arm of clinical trials and in real clinical setting not receiving any disease modifying treatment. As it was mentioned that very few patients with 4SC and NSAA

assessments over 48 weeks experienced an improvement in FMS, and such patients were excluded from the analysis. Is my understanding correct ? If so , can the authors have this clinical spelled out ?

6. PLOS authors have the option to publish the peer review history of their article (what does this mean?). If published, this will include your full peer review and any attached files.

Reviewer #1: No

Reviewer #2: No

Reviewer #3: **Yes: **Meihuan Huang

Reviewer #4: No

Reviewer #5: No

---

## [Author Response · Author response to Decision Letter 0]

23 Feb 2024

The response letter to reviewer and editor comments has been uploaded as a separate file labeled 'Response to Reviewers' as instructed.

---

## [Decision Letter · Decision Letter 1]

19 Apr 2024

PONE-D-23-24837R1Meaningful changes in motor function in Duchenne muscular dystrophy (DMD): a multi-center studyPLOS ONE

Dear Dr. Signorovitch,

Thank you for submitting your manuscript to PLOS ONE. After careful consideration, we feel that it has merit but does not fully meet PLOS ONE’s publication criteria as it currently stands. Therefore, we invite you to submit a revised version of the manuscript that addresses the points raised during the review process.

We look forward to receiving your revised manuscript.

Kind regards,

Atsushi Asakura, Ph.D

Academic Editor

PLOS ONE

Journal Requirements:

Additional Editor Comments:

Please respond to the following reviewer's comments:

The authors intended to evaluate the minimal clinically important differences to determine the minimal degree of measured change needed with > 1000 Duchenne muscular dystrophy (DMD) patients. They reported the identified thresholds for monitoring individual changes in motor function in ambulatory DMD.

1. Line 118. The patient samples were from various resources. Is there any potential heterogeneity to be concerned?

2. Line 134. “Clinical co-authors of this manuscript who cared for patients at the included RWD/NHD sources may have access to information that could identify individual participants during or after data collection.” Will their access have a potential effect on sample selection or data collection?

3. Line 174. “In RWD sources, …… may not require a patient with very poor function to attempt the test….”. This type of missing data would easily introduce bias results to be evaluated. 

Reviewers' comments:

Reviewer's Responses to Questions

**Comments to the Author**

1. If the authors have adequately addressed your comments raised in a previous round of review and you feel that this manuscript is now acceptable for publication, you may indicate that here to bypass the “Comments to the Author” section, enter your conflict of interest statement in the “Confidential to Editor” section, and submit your "Accept" recommendation.

Reviewer #2: All comments have been addressed

Reviewer #3: All comments have been addressed

Reviewer #4: All comments have been addressed

Reviewer #5: All comments have been addressed

Reviewer #6: (No Response)

2. Is the manuscript technically sound, and do the data support the conclusions?

Reviewer #2: Yes

Reviewer #3: Yes

Reviewer #4: Yes

Reviewer #5: Yes

Reviewer #6: (No Response)

3. Has the statistical analysis been performed appropriately and rigorously? 

Reviewer #2: Yes

Reviewer #3: Yes

Reviewer #4: Yes

Reviewer #5: Yes

Reviewer #6: (No Response)

4. Have the authors made all data underlying the findings in their manuscript fully available?

Reviewer #2: Yes

Reviewer #3: Yes

Reviewer #4: Yes

Reviewer #5: Yes

Reviewer #6: (No Response)

5. Is the manuscript presented in an intelligible fashion and written in standard English?

Reviewer #2: Yes

Reviewer #3: Yes

Reviewer #4: Yes

Reviewer #5: Yes

Reviewer #6: (No Response)

6. Review Comments to the Author

**Reviewer #2:** I was already happy with the first version, now the manuscript improved further following reviewing process. Perfect manuscript.

**Reviewer #3**: All my concerns have been well addressed. Great work.

**Reviewer #4:** All comments from reviewers have been answered appropriately and there are no additional points to comment.

**Reviewer #5:** Thank you for the thoroughness in explanation and have the relevant changes incorporated in the updated manuscript. The current manuscript with the updated explanation and modifications will help the readers to understand better the significance of the current findings and the clinical interpretation when applying to clinical setting. I have no further comment. Congratulation to the team for this extensive and important work.

**Reviewer #6**: The authors intended to evaluate the minimal clinically important differences to determine the minimal degree of measured change needed with > 1000 Duchenne muscular dystrophy (DMD) patients. They reported the identified thresholds for monitoring individual changes in motor function in ambulatory DMD.

1. Line 118. The patient samples were from various resources. Is there any potential heterogeneity to be concerned?

2. Line 134. “Clinical co-authors of this manuscript who cared for patients at the included RWD/NHD sources may have access to information that could identify individual participants during or after data collection.” Will their access have a potential effect on sample selection or data collection?

3. Line 174. “In RWD sources, …… may not require a patient with very poor function to attempt the test….”. This type of missing data would easily introduce bias results to be evaluated.

7. PLOS authors have the option to publish the peer review history of their article (what does this mean?). If published, this will include your full peer review and any attached files.

Reviewer #2: No

Reviewer #3: **Yes: **Meihuan Huang

Reviewer #4: No

Reviewer #5: **Yes: **Chan HS Sophelia

Reviewer #6: No

---

## [Author Response · Author response to Decision Letter 1]

13 May 2024

We have uploaded our response to reviewers letter as a Word document as instructed.

---

## [Editor Report · Decision Letter 2]

22 May 2024

Meaningful changes in motor function in Duchenne muscular dystrophy (DMD): a multi-center study

PONE-D-23-24837R2

Dear Dr. Signorovitch,

We’re pleased to inform you that your manuscript has been judged scientifically suitable for publication and will be formally accepted for publication once it meets all outstanding technical requirements.

Kind regards,

Atsushi Asakura, Ph.D

Academic Editor

PLOS ONE
---

## [Editor Report · Acceptance letter]

5 Jun 2024

PONE-D-23-24837R2 

PLOS ONE

Dear Dr. Signorovitch, 

I'm pleased to inform you that your manuscript has been deemed suitable for publication in PLOS ONE. Congratulations! Your manuscript is now being handed over to our production team.

Kind regards, 

on behalf of

Dr. Atsushi Asakura 

Academic Editor

PLOS ONE